# CEB IMPROVES MODEL ROBUSTNESS

## ABSTRACT

We demonstrate that the Conditional Entropy Bottleneck (CEB) can improve model robustness. CEB is an easy strategy to implement and works in tandem with data augmentation procedures. We report results of a large scale adversarial robustness study on CIFAR-10, as well as the IMAGENET-C Common Corruptions Benchmark, IMAGENET-A, and PGD attacks.

## 1 INTRODUCTION

We aim to make models that make meaningful predictions beyond the data they were trained on. Generally we want our models to be *robust*. Broadly, robustness is the ability of a model to continue making valid predictions as the distribution the model is tested on moves away from the empirical training set distribution. The most commonly reported robustness metric is simply test set performance, where we verify that our model continues to make valid predictions on what we hope represents valid draws from the exact same data generating procedure.

Adversarial robustness tests robustness in a worst case setting, where an attacker (Szegedy et al., 2013) makes limited targeted modifications to the input that are as fooling as possible. Many adversarial attacks have been proposed and studied (Szegedy et al., 2013; Carlini & Wagner, 2017b;a; Kurakin et al., 2016a; Madry et al., 2017). Most machine-learned systems are currently believed to be vulnerable to adversarial examples. Many defenses have been proposed, but very few have demonstrated robustness against a powerful, general-purpose adversary (Carlini & Wagner, 2017a; Athalye et al., 2018). While robustness to adversarial attacks continues to attract interest, recent discussions have emphasized the need to consider other forms of robustness as well (Engstrom et al., 2019). The Common Corruptions Benchmark (Hendrycks & Dietterich, 2019) measures image models robustness to more mild but real world sorts of perturbations. Even these modest perturbations can be very fooling for traditional architectures.

One of the few general purpose strategies that demonstrably improves model robustness is Data Augmentation (Cubuk et al., 2018; Lopes et al., 2019; Yin et al., 2019). However, it would be nice to identify loss-based solutions that can work in tandem with the data augmentation approaches. Intuitively, by performing modifications of the inputs at training time, the model is prevented from being too sensitive to particular features of the inputs that don't survive the augmentation procedure.

Alternatively, we can try to make our models more robust by making them less sensitive to the inputs in the first place. The goal of this work is to experimentally investigate whether, by systematically limiting the complexity of the extracted representation using the Conditional Entropy Bottleneck (CEB), we can make our models more robust in all three of these senses: test set generalization (e.g., classification accuracy on "clean" test inputs), worst-case robustness, and typical-case robustness.

### 1.1 CONTRIBUTIONS

This paper is primarily empirical. We demonstrate:

- CEB models are easy to implement and train.
- CEB models demonstrate improved generalization performance over deterministic baselines on CIFAR-10 and ImageNet.
- CEB models show improved robustness to adversarial attacks on CIFAR-10.
- CEB models show improved robustness on the IMAGENET-C Common Corruptions Benchmark, the IMAGENET-A Benchmark, and targeted PGD attacks.

Additionally, we show that adversarially-trained models *fail* to generalize to attacks they weren't trained on, by comparing the results on $L_2$ PGD attacks from Madry et al. (2017) to our results on the same baseline architecture. This result underscores the importance of finding ways to make models robust that do not rely on knowing the form of the attack ahead of time.

## 2 BACKGROUND

### 2.1 INFORMATION BOTTLENECKS

The Information Bottleneck (IB) objective (Tishby et al., 2000) aims to learn a stochastic representation $Z \sim p(z|x)$ that retains as much information about a target variable $Y$ while being as compressed as possible. The objective:[1]

$$\max I(Z;Y) - \sigma(-\rho)I(Z;X), \tag{1}$$

uses a Lagrange multiplier $\sigma(-\rho)$ to trade off between the relevant information ($I(Z;Y)$) and complexity of the representation ($I(Z;X)$). Because $Z$ depends only on $X$ ($Z \leftarrow X \leftrightarrow Y$): $Z$ and $Y$ are conditionally independent given $Z$:

$$I(Z;X,Y) = I(Z;X) + \underline{I(Z;Y|X)} = I(Z;Y) + I(Z;X|Y). \tag{2}$$

This allows us to write the information bottleneck of Equation (1) in an equivalent form:

$$\max I(Z;Y) - e^{-\rho}I(Z;X|Y). \tag{3}$$

Just as the original Information Bottleneck objective (Equation (1)) admits a natural variational lower bound (Alemi et al., 2017), so does this form. We can variationally lower bound the mutual information between our representation and the targets with a variational decoder $q(y|z)$:

$$I(Z;Y) = \mathbb{E}_{p(x,y)p(z|x)}\left[\log \frac{p(y|z)}{p(y)}\right] \geq H(Y) + \mathbb{E}_{p(x,y)p(z|x)}\left[\log q(y|z)\right]. \tag{4}$$

While we may not know $H(Y)$ exactly for real world datasets, in the information bottleneck formulation it is a constant outside of our control and so can be dropped in our objective. We can variationally upper bound our residual information:

$$I(Z;X|Y) = \mathbb{E}_{p(x,y)p(z|x)}\left[\log \frac{p(z|x,y)}{p(z|y)}\right] \leq \mathbb{E}_{p(x,y)p(z|x)}\left[\log \frac{p(z|x)}{q(z|y)}\right], \tag{5}$$

with a variational class conditional marginal $q(z|y)$ that approximates $\int dx\, p(z|x)p(x|y)$. Putting both bounds together gives us the Conditional Entropy Bottleneck objective (Fischer, 2018):

$$\min_{p(z|x)} \mathbb{E}_{p(x,y)p(z|x)}\left[\log q(y|z) - e^{-\rho}\log \frac{p(z|x)}{q(z|y)}\right] \tag{6}$$

Compare this with the Variational Information Bottleneck (VIB) objective (Alemi et al., 2017):

$$\min_{p(z|x)} \mathbb{E}_{p(x,y)p(z|x)}\left[\log q(y|z) - \sigma(-\rho)\log \frac{p(z|x)}{q(z)}\right]. \tag{7}$$

The difference between CEB and VIB is the presence of a class conditional versus unconditional variational marginal. As can be seen in Equation (5): using an unconditional marginal provides a looser variational upper bound on $I(Z;X|Y)$. CEB (Equation (6)) can be thought of as a tighter variational approximation than VIB (Equation (7)) to Equation (3). Since Equation (3) is equivalent to the IB objective (Equation (1)), CEB can be thought of as a tighter variational approximation to the IB objective than VIB.

---

[1] The IB objective is ordinarily written with a Lagrange multiplier $\beta \equiv \sigma(-\rho)$ with a natural range from 0 to 1. Here we use the sigmoid function: $\sigma(-\rho) \equiv \frac{1}{1+e^\rho}$ to reparameterize in terms of a control parameter $\rho$ on the whole real line. As $\rho \to \infty$ the bottleneck turns off.

## 2.2 IMPLEMENTING A CEB MODEL

In practice, turning an existing classifier architecture into a CEB model is very simple. For the stochastic representation $p(z|x)$ we simply use the original architecture, replacing the final softmax layer with a dense layer with $d$ outputs. These outputs are then used to specify the means of a $d$-dimensional Gaussian distribution with unit diagonal covariance. That is, to form the stochastic representation, independent standard normal noise is simply added to the output of the network $(z = x + \epsilon)$. For every input, this stochastic encoder will generate a random $d$-dimensional output vector. For the variational classifier $q(y|z)$ any classifier network can be used, including just a linear softmax classifier as done in these experiments. For the variational conditional marginal $q(z|y)$ it helps to use the same distribution as output by the classifier. For the simple unit variance Gaussian encoding we used in these experiments, this requires learning just $d$ parameters per class. For ease of implementation, this can be represented as single dense linear layer mapping from a one-hot representation of the labels to the $d$-dimensional output, interpreted as the mean of the corresponding class marginal.

In this setup the CEB loss takes a particularly simple form:

$$\mathbb{E}\left[w_y \cdot (f(x) + \epsilon) - \log \sum_{y'} e^{w_{y'} \cdot (f(x) + \epsilon)} - \frac{e^{-\rho}}{2}(f(x) - \mu_y)(f(x) - \mu_y + 2\epsilon)\right]. \quad (8)$$

Here the first term is the usual softmax classifier loss, but acting on our stochastic representation $z = f(x) + \epsilon$, which is simply the output of our encoder network $f(x)$ with additive Gaussian noise. The $w_y$ is the $y$th row of weights in the final linear layer outputing the logits. $\mu_y$ are the learned class conditional means for our marginal. $\epsilon$ are standard normal draws from an isotropic unit variance Gaussian with the same dimension as our encoding $f(x)$. The second term in the loss is a stochastic sampling of the KL divergence between our encoder likelihood and the class conditional marginal likelihood. $\rho$ controls the strength of the bottleneck and can vary on the whole real line. As $\rho \to \infty$ the bottleneck is turned off. In practice we find that $\rho$ values near but above 0 tend to work best for modest size models, with the tendency for the best $\rho$ to approach 0 as the model capacity increases. Notice that in expectation the second term in the loss is $(f(x) - \mu_y)^2$, which encourages the learned means $\mu_y$ to converge to the average of the representations of each element in the class. During testing we use the mean encodings and remove the stochasticity.

In its simplest form, CEB training a classifier amounts to injecting Gaussian random noise in the penultimate layer and learning estimates of the class averaged output of that layer with the stochastic regularization shown. In Appendix B we show simple modifications to the TPU-compatible ResNet implementation available on GitHub from the Google TensorFlow Team that produce the same core ResNet-50 models we use for our ImageNet experiments.

## 2.3 ADVERSARIAL ATTACKS AND DEFENSES

**Attacks.** The first adversarial attacks were proposed in Szegedy et al. (2013); Goodfellow et al. (2015). Since those seminal works, an enormous variety of attacks has been proposed (Kurakin et al. (2016a;b); Moosavi-Dezfooli et al. (2016); Carlini & Wagner (2017b); Madry et al. (2017); Eykholt et al. (2017); Baluja & Fischer (2017), etc.). In this work, we will primarily consider the Projected Gradient Descent (PGD) attack (Madry et al., 2017), which is a multi-step variant of the early Fast Gradient Method (Goodfellow et al., 2015). The attack can be viewed as having four parameters: $p$, the norm of the attack (typically 2 or $\infty$), $\epsilon$, the radius the the $p$-norm ball within which the attack is permitted to make changes to an input, $n$, the number of gradient steps the adversary is permitted to take, and $\epsilon_i$, the per-step limit to modifications of the current input. In this work, we consider $\text{L}_2$ and $\text{L}_\infty$ attacks of varying $\epsilon$ and $n$, and with $\epsilon_i = \frac{4}{3}\frac{\epsilon}{n}$.

**Defenses.** A common defense for adversarial examples is adversarial training. Adversarial training was originally proposed in Szegedy et al. (2013), but was not practical until the Fast Gradient Method was introduced. It has been studied in detail, with varied techniques (Kurakin et al., 2016b; Madry et al., 2017; Ilyas et al., 2019; Xie et al., 2019). Adversarial training can clearly be viewed as a form of data augmentation (Tsipras et al., 2018), where instead of using some fixed set of functions to modify the training examples, we use the model itself in combination with one or more

adversarial attacks to modify the training examples. As the model changes, the distribution of modifications changes as well. However, unlike with non-adversarial data augmentation techniques, such as AUTOAUG, adversarial training techniques considered in the literature so far cause substantial reductions in accuracy on clean test sets. For example, the CIFAR-10 model described in Madry et al. (2017) gets 95.5% accuracy when trained normally, but only 87.3% when trained on $L_\infty$ adversarial examples. More recently, Xie et al. (2019) adversarially trains ImageNet models with impressive robustness to targeted PGD $L_\infty$ attacks, but at only 62.32% accuracy on the non-adversarial test set, compared to 78.81% accuracy for the same model trained only on clean images.

### 2.4 COMMON CORRUPTIONS

The Common Corruptions Benchmark (Hendrycks & Dietterich, 2019) offers a real world test of model robustness in light of common image processing pipeline corruptions. Figure 4 shows examples of the 15 corruptions present in the benchmark. IMAGENET-C is a modified test set of Imagenet images with the 15 corruptions applied at five different strengths. Within each corruption type we evaluated the average error at each of the five levels ($E_c = \frac{1}{5}\sum_{s=1}^{5} E_{cs}$). To summarize the performance across all corruptions we report not only the average corruption error across all 15 tasks (avg $= \frac{1}{15}\sum_c E_c$), but also the commonly reported *Mean Corruption Error* (mCE), which reweights the errors on each task according to the performance of a baseline ALEXNET model (Hendrycks & Dietterich, 2019):

$$\text{mCE} = \frac{1}{15}\sum_c \frac{\sum_{s=1}^{5} E_{cs}}{\sum_{s=1}^{5} E_{cs}^{\text{ALEXNET}}}. \tag{9}$$

There are slightly different pipelines that have been used in the literature for the IMAGENET-C task (Lopes et al., 2019). In this work we used the ALEXNET normalization numbers and data formulation as in Yin et al. (2019).

### 2.5 NATURAL ADVERSARIAL EXAMPLES

The IMAGENET-A Benchmark (Hendrycks et al., 2019) is a dataset of 7,500 naturally-occurring "adversarial" examples across 200 ImageNet classes. The images exploit commonly-occurring weaknesses in ImageNet models, such as relying on textures often seen with certain class labels.

## 3 EXPERIMENTS

### 3.1 FASHION-MNIST EXPERIMENTS

As a warm up, we consider Wide ResNet (Zagoruyko & Komodakis, 2016) models trained on Fashion MNIST (Xiao et al., 2017), and evaluated on targeted PGD $L_2$ and $L_\infty$ attacks. All of the attacks are targeting the *trouser* class of Fashion MNIST, as that is the most distinctive class. Targeting a

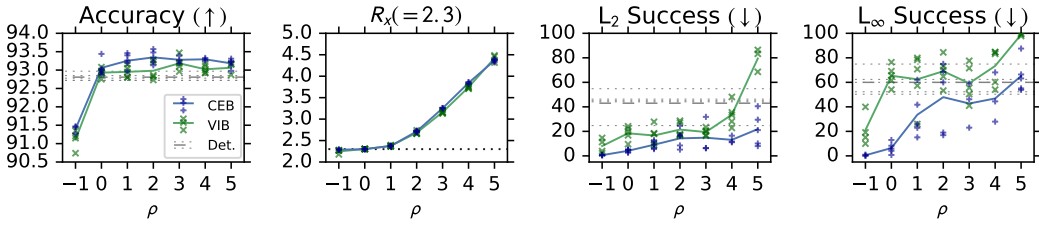

Figure 1: Clean accuracy, maximum rate lower bound $R_X \leq I(Z;X)$ seen during training, and robustness to targeted PGD $L_2$ and $L_\infty$ attacks on CEB, VIB, and Deterministic models trained on Fashion MNIST. We can see that at any given value of $\rho$, the CEB models dominate the VIB models on both accuracy and robustness, while having essentially identical maximum rates. None of these models are adversarially trained.

less distinctive class, such as one of the shirt classes, would confuse the difficulty of classifying the different shirts and the robustness of the model to adversaries. To measure robustness to the targeted attacks, we count the number of predictions that changed from a correct prediction on the clean image to an incorrect prediction of the target class on the adversarial image, and divide by the original number of correct predictions. Results are shown in Figure 1.

In this experiment we wanted to compare the performance of VIB, CEB, and a deterministic baseline. In Figure 1 (left) we see that both VIB and CEB have improved accuracy over the deterministic baseline. In order to compare the relative complexity of the learned representations for the two models, in the second panel we show the maximum lower bound seen during training on the *rate*:
$\mathbb{E}\left[\log \frac{p(z|x)}{\frac{1}{K}\sum_k^K p(z|x_k)}\right] \leq I(Z;X)$ using the encoder's minibatch marginal for both VIB and CEB.[2]
The two sets of models show nearly the same rate lower bound at each value of $\rho$.

The right two panels of Figure 1 show robustness to the targeted PGD $L_2$ and $L_\infty$ attacks. Here CEB outperforms VIB. We also see for both models that as $\rho$ decreases, the robustness to both attacks increases. In line with the proposed Minimum Necessary Information criterion from Fischer (2018), at $\rho = 0$ we end up with CEB models that have hit exactly 2.3 nats for the rate lower bound, have maintained high accuracy, and have strong robustness to both attacks. Moving to $\rho = -1$ gives only a small improvement to robustness, at the cost of a large decrease in accuracy.

## 3.2 CIFAR-10 EXPERIMENTS

**28×10 Wide ResNet Experiments**  We trained a set of 25 28×10 Wide ResNet (WRN) CEB models on CIFAR-10 at $\rho \in [-1, -0.75, ..., 5]$, as well as a deterministic baseline. They trained for 1500 epochs, lowering the learning rate by a factor of 0.3 after 500, 1000, and 1250 epochs. This long training regime was due to our use of the original AUTOAUG policies, which requires longer training. The only additional modification we made to the basic 28×10 WRN architecture was the removal of all Batch Normalization (Ioffe & Szegedy, 2015) layers. Every small CIFAR-10 model we have trained with Batch Normalization enabled has had substantially worse robustness to $L_\infty$ PGD adversaries, even though typically the accuracy is much higher. For example, 28×10 WRN CEB models rarely exceeded more than 10% adversarial accuracy. However, it was always still the case that lower values of $\rho$ gave higher robustness. As a baseline comparison, a deterministic 28×10 WRN with BatchNorm, trained with AUTOAUG reaches 97.3% accuracy on clean images, but 0% accuracy on $L_\infty$ PGD attacks at $\epsilon = 8$ and $n = 20$. Interestingly, that model was noticeably more robust to $L_2$ PGD attacks than the deterministic baseline without BatchNorm, getting 73% accuracy compared to 66%. However, it was still much weaker than the CEB models, which get over 80% accuracy on the same attack (Figure 2). Additional training details are in Appendix A.1.

Figure 2 demonstrates the adversarial robustness of CEB models to both targeted $L_2$ and $L_\infty$ attacks. The CEB models show a marked improvement in robustness to $L_2$ attacks compared to an adversarially-trained baseline from Madry et al. (2017) (denoted Madry). Figure 3 shows the robustness of five of those models to PGD attacks as $\epsilon$ is varied. We selected the four CEB models to represent the most robust models across most of the range of $\rho$ we trained. Note that of the 25 CEB models we trained, only the models with $\rho \geq 1$ succesfully trained. The remainder collapsed to chance performance. This is something we observe on all datasets when training models that are too low capacity. Only by increasing model capacity does it become possible to train at low $\rho$. Note that this result is predicted by the theory of the onset of learning in the Information Bottleneck and its relationship to model capacity from Wu et al. (2019).

**62×7 Wide ResNet Experiments.**  In order to explore the effect of model size on training, and to train at lower $\rho$, we trained the largest Wide ResNet we could fit on a single GPU with a batch size of 250. This was a 62×7 model similar to the ones above, including the use of AUTOAUG, but we additionally enabled BatchNorm. We were able to train at $\rho = 0$ with this larger model, which reached **97.51%** accuracy. This result is better than the 28×10 Wide ResNet from AUTOAUG by 0.19 percentage points, although it is still worse than the Shake-Drop model from that paper. We additionally tested the model on the new CIFAR-10.1 test set (Recht et al., 2018), getting accuracy of 93.6%. This is a gap of **3.9** percentage points, which is better than all of the results reported in that paper, and substantially better than the Wide ResNet results (but still inferior to the Shake-Drop

---

[2]This lower bound on $I(X;Z)$ is the "InfoNCE with a tractable encoder" bound from Poole et al. (2019).

AUTOAUG results). The same model at $\rho = 5$ reached 97.05% accuracy on the normal test set and 91.9% on the CIFAR-10.1 test set, showing that increased $\rho$ gave substantially worse generalization.

To test robustness of these models, we swept $\epsilon$ for both PGD attacks, which we show in Figure 3. The main result is that the $62 \times 7$ CEB$_0$ model not only has substantially higher accuracy than baseline Wide ResNets trained with AUTOAUG, it also beats the adversarially-trained model on both the L$_2$ and the L$_\infty$ attacks at almost all values of $\epsilon$. We also show that this model is even more robust to two transfer attacks, where we used the $62 \times 7$ CEB$_5$ model and the adversarially-trained model to generate PGD attacks, and then test them on the CEB$_0$ model. This result helps to counter possible claims that these models are doing "gradient masking" (the more compelling evidence against gradient masking is that the robustness of the model is strongly correlated with the hyperparameter $\rho$, whose only effect is to constrain the amount of information the model captures).

We additionally tested both models on the CIFAR-10 Common Corruptions test sets. At the time of training, we were unaware that AUTOAUG's default policies for CIFAR-10 contain brightness and contrast augmentations that amount to training on those two corruptions from Common Corruptions (as mentioned in Yin et al. (2019)), so our results are not appropriate for direct comparison with other results in the literature. However, they still allow us to compare the effect of bottlenecking the information between these two large models. The $\rho = 5$ model reached an mCE[3] of 61.2. The $\rho = 0$ model reached an mCE of 52.0, which is a dramatic relative improvement.

### 3.3 IMAGENET EXPERIMENTS

To demonstrate CEB's ability to improve robustness to real world data shifts, we trained six different types of networks on ImageNet at $224 \times 224$ resolution and two different sizes of RESNET, RESNET-50 and RESNET-152, and then tested them on IMAGENET-C, IMAGENET-A, and targeted PGD attacks. As a simple baseline we trained RESNET-50 networks with no data augmentation. We then trained the same networks but as CEB networks at ten different values of $\rho = (1, 2, \ldots, 10)$. AUTOAUG (Cubuk et al., 2018) has previously been demonstrated to improve robustness markedly on IMAGENET-C so next we trained our baseline RESNET-50 model with AUTOAUG. We similarly trained these AUTOAUG models as CEB models with ten different values of $\rho$. IMAGENET-C numbers are also sensitive to the model capacity. To assess whether CEB can benefit larger models we repeated the experiments with a modified RESNET-50 network where every layer was made twice as wide. Finally, we repeated the above six model types with RESNET-152 baselines and CEB models without AUTOAUG, with AUTOAUG, and with AUTOAUG and twice as wide. All other hyperparameters (learning rate schedule, L$_2$ weight decay scale, etc.) remained the same across all models. In total we trained 66 ImageNet models – 6 deterministic baselines varying augmentation, width, and depth, and 60 CEB models additionally varying $\rho$. The results for the RESNET-50 models are summarized in Figure 4 and Table 1. For RESNET-152, see Figure 5 and Table 2.

The CEB models highlighted in Figures 4 and 5 and Tables 1 and 2 were selected by cross validation. These were values of $\rho$ that gave the best *clean* test set accuracy. Despite being selected for classical generalization, these models also demonstrate a high degree of robustness on both average- and worst-case perturbations. In the case that more than one model gets the same test set accuracy, we choose the model with the lower $\rho$, since we know that lower $\rho$ correlates with higher robustness. The only model where we had to make this decision was for RESNET-152 with AUTOAUG, where five models all were within 0.1% of each other, so we chose the $\rho = 3$ model, rather than $\rho \in \{5...8\}$.

**IMAGENET-C and IMAGENET-A.** Both data augmentation and increasing model capacity have positive effects on robustness to both IMAGENET-C and IMAGENET-A, but for all three classes of models CEB gives substantial additional improvements.

**Targeted PGD Attacks.** We tested on the random-target version of the PGD L$_2$ and L$_\infty$ attacks (Kurakin et al., 2016a), both at $\epsilon = 16$, $n = 20$, and $\epsilon_i = 2$, which is considered to be a strong attack still (Xie et al., 2019). Similar to our results on CIFAR-10, model capacity makes a substantial difference to whitebox adversarial attacks. In particular, none of the ResNet-50 models perform well, getting less than 1% top-1 accuracy. However, the Resnet-152 CEB models show a dramatic improvement over the deterministic baseline models, with top-1 accuracy increasing from

---

[3]The mCE is computed relative to a baseline model. We use the baseline model from Yin et al. (2019).

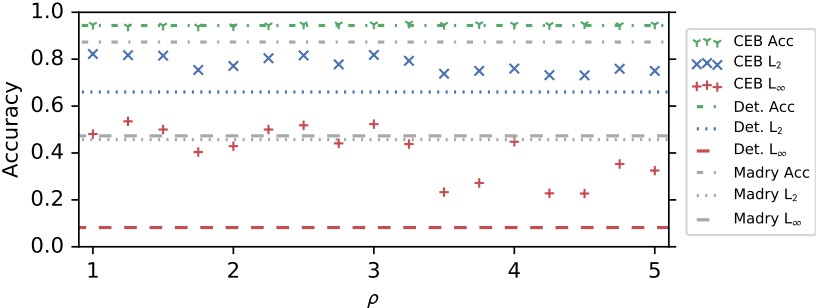

Figure 2: CEB $\rho$ vs. test set accuracy, and $L_2$ and $L_\infty$ PGD adversarial attacks on CIFAR-10. The attack parameters were selected to be about equally difficult for the adversarially-trained WRN 28×10 model from Madry et al. (2017) (grey dashed and dotted lines). The deterministic baseline (Det.) only gets 8% accuracy on the $L_\infty$ attacks, but gets 66% on the $L_2$ attack, substantially better than the 45.7% of the adversarially-trained model, which makes it clear that the adversarially-trained model failed to generalize in any reasonable way to the $L_2$ attack. The CEB models are always substantially more robust than Det., and many of them outperform Madry even on the $L_\infty$ attack the Madry model was trained on, but for both attacks there is a clear general trend toward more robustness as $\rho$ decreases. Finally, the CEB and Det. models all reach about the same accuracy, ranging from 93.9% to 95.1%, with Det. at 94.4%. In comparison, Madry only gets 87.3%. We emphasize that none of the CEB models is adversarially trained.

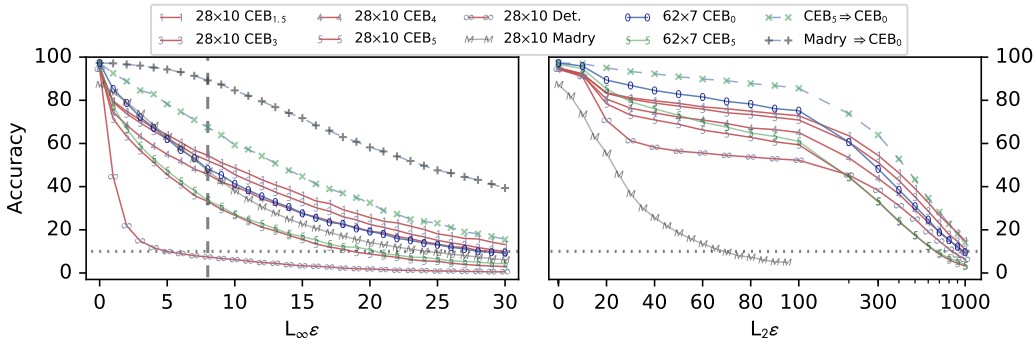

Figure 3: Untargeted adversarial attacks on small and large CIFAR-10 models showing both strong robustness to PGD $L_2$ and $L_\infty$ attacks, as well as excellent test accuracy of 97.5%. **Left:** Accuracy on untargeted $L_\infty$ attacks at different values of $\varepsilon$ for all 10,000 test set examples. 28×10 and 62×7 indicate the Wide ResNet size. $CEB_x$ indicates a CEB model trained at $\rho = x$. Madry is the adversarially-trained model from Madry et al. (2017) (values provided by Aleksander Madry). Madry was trained with 7 steps of $L_\infty$ PGD at $\varepsilon = 8$ (grey dashed line). 62×7 $CEB_0$ is the CEB model with the highest accuracy (97.51%) trained at $\rho = 0$. 62×7 $CEB_5$ is the CEB model with the highest accuracy (97.05%) trained at $\rho = 5$. $CEB_5 \Rightarrow CEB_0$ is transfer attacks from the 62×7 $CEB_5$ model to the 62×7 $CEB_0$ model. Madry $\Rightarrow CEB_0$ is transfer attacks from the Madry model to the 62×7 $CEB_0$ model. All of the CEB models with $\rho \leq 4$ outperform Madry across most of the values of $\epsilon$, even though they were not adversarially-trained. The 62×7 $CEB_0$ model is even more robust to transfer attacks than to the direct whitebox attacks. **Right:** Accuracy on untargeted $L_2$ attacks at different values of $\varepsilon$. Note the switch to log scale on the x axis at $L_2\epsilon = 100$. All values are collected at 20 steps of PGD. It is interesting to note that both the small and the large $CEB_5$ models have essentially identical robustness, in spite of very different numbers of parameters, and that the Det. model eventually outperforms the $CEB_5$ models on $L_2$ attacks at relatively high accuracies. We emphasize that none of the CEB models is adversarially trained.

| type | CEBx2 | 50x2 | CEB | 50 | CEB-aa | 50-aa |
|---|---|---|---|---|---|---|
| $\rho^*$ | 2 | NA | 5 | NA | 6 | NA |
| Clean | **20.5%** | 21.8% | **22.3%** | 22.5% | **23.0%** | 24.0% |
| mCE | **65.6%** | 69.7% | **69.9%** | 72.2% | **77.7%** | 81.2% |
| Average CE | **51.9%** | 55.2% | **55.3%** | 57.2% | **61.5%** | 64.3% |
| Gaussian Noise | **53.1%** | 57.2% | **57.4%** | 59.8% | **66.0%** | 71.3% |
| Shot Noise | **53.8%** | 57.9% | **58.0%** | 60.2% | **68.0%** | 72.8% |
| Impulse Noise | **59.4%** | 63.8% | **63.1%** | 66.4% | **71.4%** | 76.8% |
| Defocus Blur | **60.0%** | 61.9% | **63.5%** | 64.4% | **66.0%** | 66.3% |
| Glass Blur | **63.8%** | 65.0% | **66.1%** | 67.1% | **74.0%** | 74.5% |
| Motion Blur | **57.1%** | 60.6% | **60.4%** | 63.2% | **64.2%** | 66.3% |
| Zoom Blur | **59.2%** | 62.1% | **62.6%** | 65.2% | **64.1%** | 64.5% |
| Snow | **61.1%** | 65.8% | **64.8%** | 68.3% | **71.0%** | 74.5% |
| Frost | **56.3%** | 59.5% | **59.9%** | 62.0% | **66.0%** | 68.3% |
| Fog | **47.4%** | 52.8% | **50.2%** | 53.0% | **60.1%** | 63.3% |
| Brightness | **28.8%** | 31.1% | **30.7%** | 31.8% | **34.7%** | 36.4% |
| Contrast | **52.5%** | 55.6% | **56.8%** | 57.3% | **65.8%** | 68.4% |
| Elastic Transform | **51.6%** | 54.0% | **54.8%** | 55.8% | **56.8%** | 57.8% |
| Pixelate | **36.2%** | 38.2% | **39.2%** | 40.6% | **49.8%** | 55.7% |
| Jpeg Compression | **38.8%** | 41.7% | **41.9%** | 42.3% | **45.0%** | 47.9% |
| ImageNet-A | **89.0%** | 90.5% | **92.5%** | 93.3% | **95.1%** | 96.8% |
| PGD $L_2$ | 99.4% | 99.9% | 99.6% | 99.9% | 99.7% | 99.9% |
| PGD $L_\infty$ | 99.4% | 99.9% | 99.6% | 99.9% | 99.7% | 99.9% |

Table 1: Baseline and cross-validated CEB values for the ImageNet experiments. The middle columns show a RESNET-50 model trained with AUTOAUG ("50") versus the corresponding CEB network ("CEB"). The right columns ("-aa") remove AUTOAUG from the data processing pipeline, and the left columns ("x2") are models that are twice as wide. The CEB values reported here are denoted with the dots in Figure 4. Lower values are better in all cases, and the lowest value in pairs of columns is shown in bold. Table 2 gives the same results for the RESNET-152 models.

0.09% to 17.09% between the deterministic baseline and the $\rho = 1$ models without AUTOAUG, a relative increase of 187 times, and increases nearly as large for the AUTOAUG and wide AUTOAUG models. Interestingly, for the PGD attacks, AUTOAUG was detrimental – the RESNET-152 models without AUTOAUG were more robust than those with AUTOAUG. Only the wide RESNET-152 models with AUTOAUG exceeded the robustness of the narrow RESNET-152 without AUTOAUG.

# 4 CONCLUSION

The Conditional Entropy Bottleneck (CEB) provides a simple mechanism to improve robustness of image classifiers. We have shown that CEB gives a tighter variational bound on the IB objective than the closely-related VIB, while also having consistently better test accuracy and robustness. We have shown a strong trend toward increased robustness as $\rho$ decreases in the standard $28 \times 10$ Wide RESNET model on CIFAR-10, and that this increased robustness does not come at the expense of accuracy relative to the deterministic baseline. We have shown that CEB models at a range of $\rho$ essentially dominate an adversarially-trained baseline model, even on the attack the adversarial model was trained on, and have incidentally shown that the adversarially-trained model generalizes to at least one other attack *less well* than a deterministic baseline. Finally, we have shown that on ImageNet, CEB provides substantial gains over deterministic baselines in both validation accuracy and robustness to Common Corruptions, Natural Adversarial Examples, and targeted Projected Gradient Descent attacks. We hope these empirical demonstrations inspire further theoretical and practical study of the use of bottlenecking techniques to encourage improvements to both classical generalization and robustness.

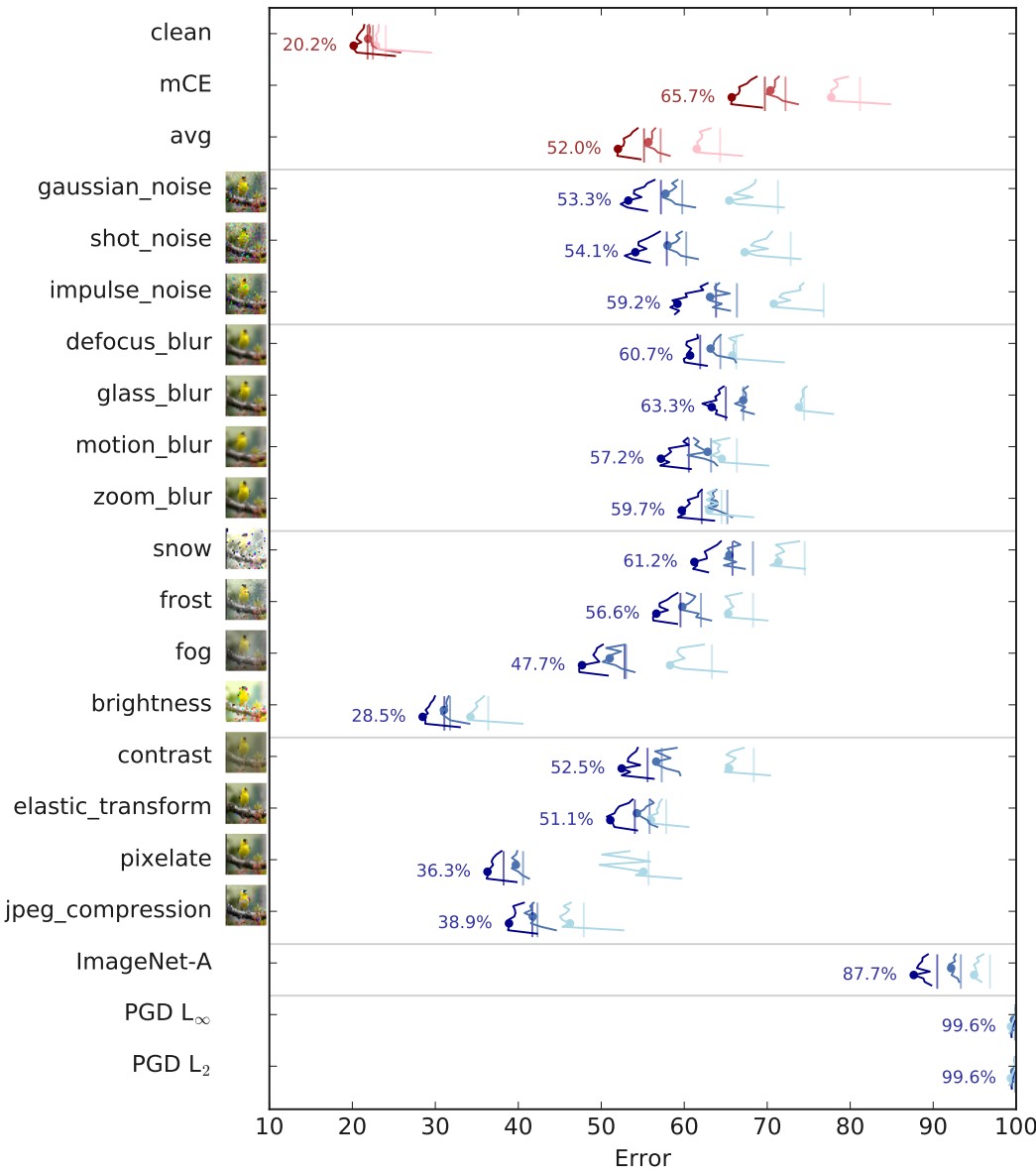

Figure 4: Summary of the IMAGENET-C experiments. In the main part of the figure (in blue), the average errors (lower is better) across corruption magnitude are shown for 33 different networks for each of the labeled Common Corruptions (Hendrycks & Dietterich, 2019), IMAGENET-A (Hendrycks et al., 2019), and targeted PGD attacks (Madry et al., 2017). The networks come in pairs, with the vertical lines denoting the baseline network's performance and then in the corresponding color the errors for each of 10 different CEB trained networks are shown with varying $\rho = [1, 2, \ldots, 10]$, arranged from 10 at the top to 1 at the bottom. The light blue lines denote a baseline RESNET-50 model trained without AUTOAUG. The blue lines show the same network (and baseline) but with AUTOAUG. The dark blue lines show RESNET-50 AUTOAUG networks that were made twice as wide. In this way we can separately see the effect of both data augmentation and enlarging the model, as well as the additive effect of CEB on each model. At the top in red are shown the same data for three summary statistics. clean denotes the clean errors of each of the networks. mCE denotes the ALEXNET regularized average corruption errors. avg shows an equally weighted average across all common corruptions. The dots denote the value for each CEB network and each corruption at $\rho^*$, the optimum $\rho$ for the network as measured in terms of clean accuracy. The values at these dots and the baseline values are given in detail in Table 1. Figure 5 and Table 2 give the same data for the RESNET-152 models.

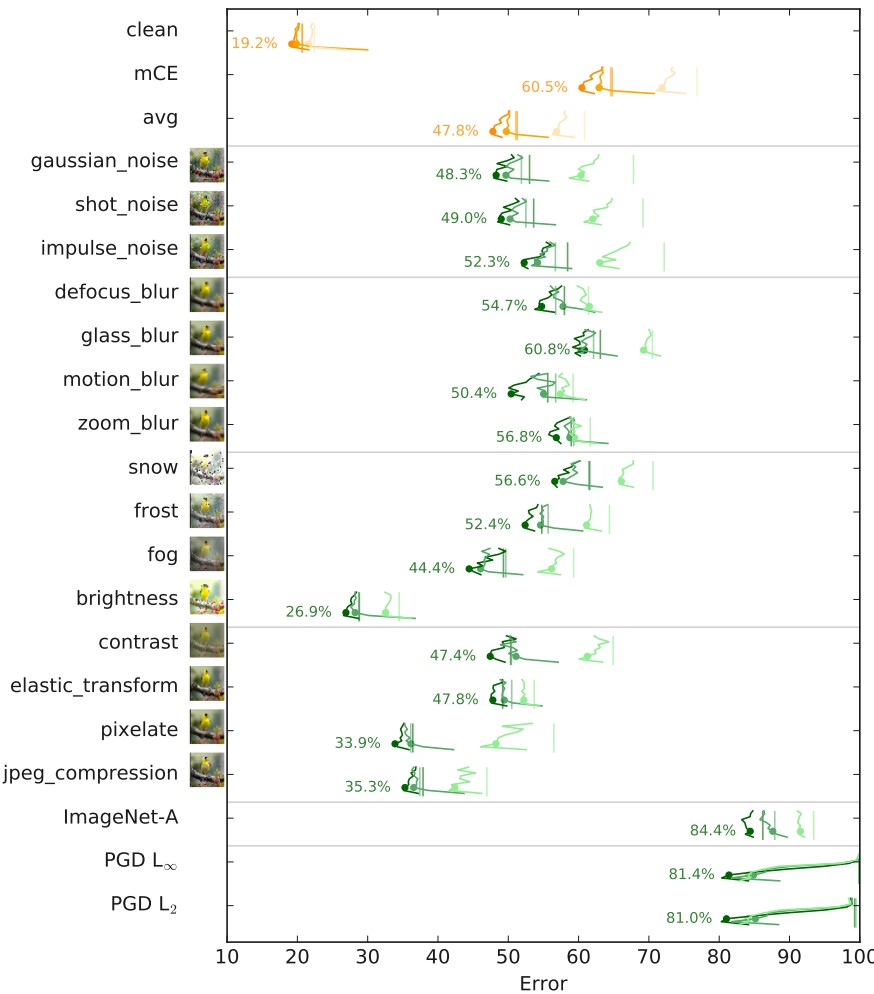

Figure 5: Replication of Figure 4 but for RESNET-152. The deeper model shows marked improvement across the board. Notice in particular the adversarial robustness to $L_\infty$ and $L_2$ PGD attacks for the CEB models over the deterministic baselines.

| type | CEB-152x2 | 152x2 | CEB-152 | 152 | CEB-152-aa | 152-aa |
|---|---|---|---|---|---|---|
| $\rho^*$ | 3 | NA | 3 | NA | 3 | NA |
| Clean | **19.2%** | 20.7% | **19.9%** | 20.7% | **21.6%** | 22.4% |
| mCE | **60.5%** | 64.8% | **63.0%** | 64.6% | **71.9%** | 76.9% |
| Average CE | **47.8%** | 51.3% | **49.7%** | 51.1% | **56.9%** | 60.8% |
| Gaussian Noise | **48.3%** | 53.0% | **49.7%** | 51.9% | **60.3%** | 67.8% |
| Shot Noise | **49.0%** | 53.6% | **50.3%** | 52.5% | **62.0%** | 69.2% |
| Impulse Noise | **52.3%** | 58.4% | **54.2%** | 56.7% | **63.0%** | 72.2% |
| Defocus Blur | **54.7%** | 58.0% | **57.8%** | 56.7% | 61.5% | **61.4%** |
| Glass Blur | **60.8%** | 63.1% | **60.3%** | 62.1% | **69.2%** | 70.5% |
| Motion Blur | **50.4%** | 55.6% | **55.0%** | 56.7% | **57.4%** | 59.2% |
| Zoom Blur | **56.8%** | 59.0% | **58.8%** | 59.3% | **59.4%** | 61.7% |
| Snow | **56.6%** | 61.6% | **57.8%** | 61.5% | **66.1%** | 70.6% |
| Frost | **52.4%** | 54.8% | **54.6%** | 55.7% | **61.1%** | 64.4% |
| Fog | **44.4%** | 49.3% | **46.0%** | 49.6% | **56.2%** | 59.3% |
| Brightness | **26.9%** | 28.8% | **28.2%** | 28.7% | **32.6%** | 34.5% |
| Contrast | **47.4%** | 50.3% | 51.1% | **50.5%** | **61.3%** | 64.9% |
| Elastic Transform | **47.8%** | 49.2% | **49.4%** | 50.5% | **52.2%** | 53.7% |
| Pixelate | **33.9%** | 36.4% | **36.1%** | **36.1%** | **48.2%** | 56.5% |
| Jpeg Compression | **35.3%** | 37.9% | **36.5%** | 37.4% | **42.4%** | 47.0% |
| ImageNet-A | **84.4%** | 86.2% | **87.6%** | 87.9% | **91.6%** | 93.4% |
| PGD $L_2$ | **81.4%** | 99.9% | **84.9%** | 99.9% | **84.2%** | 99.9% |
| PGD $L_\infty$ | **81.0%** | 99.3% | **85.1%** | 99.4% | **84.1%** | 99.5% |

Table 2: Same as Table 1 above but for a deeper RESNET-152 model. Note that for the baseline 152x2 and 152 models, the smaller model (152) actually has better mCE and equally good top-1 accuracy, indicating that the wider model may be overfitting, but the CEB-152x2 model substantially outperforms both of them across the board.

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

## A    EXPERIMENT DETAILS

Here we present additional technical details for the CIFAR-10 and ImageNet experiments.

### A.1    CIFAR-10 EXPERIMENT DETAILS

We trained all of the models using Adam (Kingma & Ba, 2015) at a base learning rate of $10^-3$. We lowered the learning rate three times by a factor of 0.3 each time. The only additional trick to train the CIFAR-10 models was to start with $\rho = 100$, anneal down to $\rho = 10$ over 2 epochs, and then anneal to the target $\rho$ over one epoch once training exceeded a threshold of 20%. This jump-start method is inspired by experiments on VIB in Wu et al. (2019). It makes it much easier to train models at low $\rho$, and appears to not negatively impact final performance.

For the 62×7 models, we used the data augmentation policies for CIFAR-10 found by AUTOAUG and trained the models for 800 epochs, lowering the learning rate by a factor of 10 at 400 and 600 epochs.

### A.2    IMAGENET EXPERIMENT DETAILS

We follow the learning rate schedule for the ResNet 50 from Cubuk et al. (2018), which has a top learning rate of 1.6, trains for 270 epochs, and drops the learning rate by a factor of 10 at 90, 180, and 240 epochs. The only difference for all of our models is that we train at a batch size of 8192 rather than 4096. Similar to the CIFAR-10 models, in order to ensure that the ImageNet models train at low $\rho$, we employ a simple jump-start. We start at $\rho = 100$ and anneal down to the target $\rho$ over 12,000 steps. The first learning rate drop occurs a bit after 14,000 steps. Also similar to the CIFAR 28×10 WRN experiments, none of the models we trained at $\rho = 0$ succeeded, indicating that RESNET-50 and WRN 50×2 both have insufficient capacity to fully learn ImageNet. We were able to train RESNET-152 at $\rho = 0$, but only by disabling $L_2$ weight decay and using a slightly lower learning rate. Since that involved additional hyperparameter tuning, we don't report those results here, beyond noting that it is possible, and that those models reached top-1 accuracy around 72%.

# B CEB EXAMPLE CODE

Here we give the core changes needed to make ResNet CEB models, based on the TPU-compatible ResNet implementation from the Google TensorFlow Team.

```python
# In model.py:
def resnet_v1_generator(block_fn, layers, num_classes, ...):
  def model(inputs, is_training):
    # Build the ResNet model as normal up to the following lines:
    inputs = tf.reshape(
        inputs, [-1, 2048 if block_fn is bottleneck_block else 512])
    # Now, instead of the final dense layer, just return inputs,
    # which for ResNet50 models is a [batch_size, 2048] tensor.
    return inputs
```

Listing 1: Modifications to the `model.py` function.

```python
# In resnet_main.py add the following imports and functions:
import tensorflow_probability as tfp
tfd = tfp.distributions

def ezx_dist(x):
  """Builds the encoder distribution, e(z|x)."""
  dist = tfd.MultivariateNormalDiag(loc=x)
  return dist

def bzy_dist(y, num_classes=1000, z_dims=2048):
  """Builds the backwards distribution, b(z|y)."""
  y_onehot = tf.one_hot(y, num_classes)
  mus = tf.layers.dense(y_onehot, z_dims, activation=None)
  dist = tfd.MultivariateNormalDiag(loc=mus)
  return dist

def cyz_dist(z, num_classes=1000):
  """Builds the classifier distribution, c(y|z)."""
  # For the classifier, we are using exactly the same dense layer
  # initialization as was used for the final layer that we removed
  # from model.py.
  logits = tf.layers.dense(
      z, num_classes, activation=None,
      kernel_initializer=tf.random_normal_initializer(stddev=.01))
  return tfd.Categorical(logits=logits)

def lerp(global_step, start_step, end_step, start_val, end_val):
  """Utility function to linearly interpolate two values."""
  interp = (tf.cast(global_step - start_step, tf.float32)
            / tf.cast(end_step - start_step, tf.float32))
  interp = tf.maximum(0.0, tf.minimum(1.0, interp))
  return start_val * (1.0 - interp) + end_val * interp
```

Listing 2: Modification to the head of `resnet_main.py`.

```python
# Still in resnet_main.py, modify resnet_model_fn as follows:
def resnet_model_fn(features, labels, mode, params):
  # Nothing changes until after the definition of build_network:
  def build_network():
    # Elided, unchanged implementation of build_network.

  if params['precision'] == 'bfloat16':
    # build_network now returns the pre-logits, so we'll change
    # the variable name from logits to net.
    with tf.contrib.tpu.bfloat16_scope():
      net = build_network()
    net = tf.cast(net, tf.float32)
  elif params['precision'] == 'float32':
    net = build_network()

  # Get the encoder, e(z|x):
  with tf.variable_scope('ezx', reuse=tf.AUTO_REUSE):
    ezx = ezx_dist(net)
  # Get the backwards encoder, b(z|y):
  with tf.variable_scope('bzy', reuse=tf.AUTO_REUSE):
    bzy = bzy_dist(labels)

  # Only sample z during training. Otherwise, just pass through
  # the mean value of the encoder.
  if mode == tf.estimator.ModeKeys.TRAIN:
    z = ezx.sample()
  else:
    z = ezx.mean()

  # Get the classifier, c(y|z):
  with tf.variable_scope('cyz', reuse=tf.AUTO_REUSE):
    cyz = cyz_dist(z, params)

  # cyz.logits is the same as what the unmodified ResNet model
  # would return.
  logits = cyz.logits

  # Compute the individual conditional entropies:
  hzx = -ezx.log_prob(z)   # H(Z|X)
  hzy = -bzy.log_prob(z)   # H(Z|Y) (upper bound)
  hyz = -cyz.log_prob(labels)  # H(Y|Z) (upper bound)

  # I(X;Z|Y) = -H(Z|X) + H(Z|Y)
  #          >= -hzx + hzy =: Rex, the residual information.
  rex = -hzx + hzy

  rho = 3.0  # You should make this a hyperparameter.
  rho_to_gamma = lambda rho: 1.0 / np.exp(rho)
  gamma = tf.cast(rho_to_gamma(rho), tf.float32)

  # Get the global step now, so that we can adjust rho dynamically.
  global_step = tf.train.get_global_step()

  anneal_rho = 12000  # You should make this a hyperparameter.
  if anneal_rho > 0:
    # Anneal rho from 100 down to the target rho
    # over the first anneal_rho steps.
    gamma = lerp(global_step, 0, aneal_rho,
                 rho_to_gamma(100.0), gamma)

  # Replace all the softmax cross-entropy loss computation with
  # the following line:
  loss = tf.reduce_mean(gamma * rex + hyz)
  # The rest of resnet_model_fn can remain unchanged.
```

Listing 3: Modifications to resnet_model_fn in resnet_main.py.

