# OpenReview forum: "CEB Improves Model Robustness"
_ICLR.cc/2020/Conference — Reject_

### Official Review · AnonReviewer2 · 2019-10-21
**Official Blind Review #2**

**Rating:** 6

**Review:**

TITLE
CEB Improves Model Robustness

REVIEW SUMMARY
As a thorough empirical evaluation of a new method for training deep neural networks, this paper is a useful contribution.

PAPER SUMMARY
The paper presents an empirical study of the robustness of neural network classifiers trained using a conditional entropy bottleneck regularization.

CLARITY
The paper is well written and easy to follow. Results are presented in a way that gives the reader a good overview, but figure quality / legibility could be improved.

ORIGINALITY
The CEB method is an original (but minor) modification of a well known method.

SIGNIFICANCE
The primary contribution of the paper is a thorough experimental evaluation of the particular regularization method presented in the paper. While this is certainly useful, it would have been even better with more comparison with other regularization methods. The paper clearly demonstrates the benefits of CEB, which is of interest in itself.

FURTHER COMMENTS
Consider making a more clear distinction between generalization and robustness.
"are are" Double word.
"all machine-learned systems...highly vunerable" Too strong statement.
"VIB" Abbreviation not defined in the text.
Figure 1, 2 and 3 are a bit hard to read (especially the dotted lines)



**Experience Assessment:**

I have read many papers in this area.

**Review Assessment: Checking Correctness Of Derivations And Theory:**

I assessed the sensibility of the derivations and theory.

**Review Assessment: Checking Correctness Of Experiments:**

I assessed the sensibility of the experiments.

**Review Assessment: Thoroughness In Paper Reading:**

I read the paper at least twice and used my best judgement in assessing the paper.

---

> ### Author Response · Authors · 2019-11-15
> **Response to Reviewer2**
>
> Thank you for your review. We have addressed your minor comments in the updated text. We agree that the early figures are difficult to read, and have improved them in the updated version. We appreciate that you found the presentation clear, the approach original, and the benefit of CEB to model robustness of interest in itself. We agree that a careful comparison of the effects of regularization techniques on robustness would be interesting for future work. We note that we implicitly have some comparisons in the paper, in that the ImageNet models are all trained with both L$_2$ weight normalization and BatchNorm, neither of which confers substantial robustness to the average case, and essentially no robustness to the worse case, when used without CEB.

---

### Official Review · AnonReviewer3 · 2019-10-22
**Official Blind Review #3**

**Rating:** 3

**Review:**

The paper modifies existing classifier architectures and training objective, in order to minimize "conditional entropy bottleneck" (CEB) objective, in attempts to force the representation to maximize the information bottleneck objective. Consequently, the paper claims that this CEB model improves general test accuracy and robustness against adversarial attacks and common corruptions, compared to the softmax + cross entropy counterpart. This claim is supported by experimental results on CIFAR-10 and ImageNet-C datasets.

In overall, the manuscript is easy-to-follow with a clear motivation. I found the experimental results are also promising, at least for the improved test accuracy and corruption robustness. Regarding the results about adversarial robustness, however, it was really confusing for me to understand and validate the reported values. I would like to increase my score if the following questions could be addressed:

- It is not clear whether adversarial training is used or not in CEB models for the adversarial robustness results. If the results were achieved "without" adversarial training, these would be somewhat surprising for me. At the same time, however, I would want to see more thorough evaluation than the current, e.g. PGD with more n, random restart, gradient-free attacks, or black-box attacks.
- I wonder if the paper could provide a motivation on why the AutoAug policy is adopted when training robust model. Personally, this is one of the reasons that makes me hard to understand the values presented in the paper.
- Figure 3, right: Does this plot indicates that "28x10 Det" is much more robust than "28x10 Madry"? If so, it feels so awkward for me, and I hope this results could be further justified in advance.
- Figure 3: It is extremely difficult to understand the plots as the whole lines are interleaved on a single grid. I suggests to split the plots based on the main claims the paper want to demonstrate.
- How was the issue of potential over-fitting on rho handled, e.g. using a validation set?
- In general, I slightly feel a lack of justification on why the CEB model improves robustness. In Page 5: "... Every small model we have trained with Batch Normalization enabled has had substantially worse robustness, ..." - I think this line could be a critical point, and need further investigations in a manner of justifying the overall claim.

**Experience Assessment:**

I have read many papers in this area.

**Review Assessment: Checking Correctness Of Derivations And Theory:**

I assessed the sensibility of the derivations and theory.

**Review Assessment: Checking Correctness Of Experiments:**

I carefully checked the experiments.

**Review Assessment: Thoroughness In Paper Reading:**

I read the paper at least twice and used my best judgement in assessing the paper.

---

> ### Author Response · Authors · 2019-11-15
> **Response to Reviewer3**
>
> Thank you for your review. We address your major concerns below.
>
> Adversarial Training.
> None of our models is adversarially trained. Only the Madry model (which we compare to as a baseline for robustness on CIFAR10) is adversarially trained. We agree that this is a surprising result, and hope that other researchers will find it as exciting as we do, since we additionally demonstrated that CEB improves clean test accuracy as well, while the adversarial training techniques proposed so far are all harmful to clean test accuracy.
>
> Adversarially-Trained Robustness to L$_2$ Attacks.
> Yes, the result is correct. The baseline deterministic model is **substantially more** robust to the PGD L$_2$ attack than the adversarially-trained model from Madry (2017). We conferred with Madry extensively to ensure that we were measuring everything in exactly the same way for Figure 3. While we did not write this paper to show that adversarial training fails to generalize to other attacks, we consider this to be a very important result for the research community on adversarial robustness. We discuss that fact in the caption for Figure 2 (which also shows the same result) and in the Conclusion. Based on your suggestion, we additionally are mentioning this result in the Introduction.
>
> Alternative Attacks.
> The transfer attacks shown for the large CEB model is an example of a blackbox attack. They show that the large CEB model is even more robust to PGD attacks generated by both the less robust large CEB model and the adversarially trained model from Madry (2017). We may have time to add other attacks to the appendix if the paper is accepted, but we note that PGD is the standard attack used in the literature, due to its flexibility, relative speed, and effectiveness. We note here that preliminarily we have tested the large CIFAR10 CEB models on all of the attacks in CleverHans. CEB helped in all instances, but we chose to focus on the PGD style attacks for the paper to try to tell a compelling story.
>
> Training with AutoAugment.
> We mention in the Introduction that good data augmentation policies are the one currently-known way to improve average-case robustness (i.e., robustness to perturbations like those found in the Common Corruptions benchmark). This is the core justification for the use of AutoAugment. There are additional pragmatic reasons for its use. For example, for CIFAR10 models, some augmentation policy is required to get good accuracy for all models we are aware of. The standard set of augmentations was shown to be inferior to the learned policies from AutoAugment, so there seems to be little point in using the standard augmentations. Furthermore, we are comparing with adversarial training, which (as we mention in the paper) can be viewed as a particular choice of data augmentation policy. Thus, comparing to models trained with AutoAugment seems reasonable. Finally, we wanted to demonstrate (with the large CEB models) that it was possible to exceed the previous SOTA accuracy on CIFAR10 for a WideResNet model that was presented in the AutoAugment paper. A core point of the experiments is to show in multiple settings that CEB not only improves the robustness of the model, it also improves classical generalization performance. Outperforming the previous SOTA results makes that claim much stronger. We discuss those comparisons in detail in Section 3.2.
>
> Clarity of Figure 3.
> We agree that it is difficult to interpret Figure 3 in the current version. We have improved it in the new version and will continue to work on its legibility for the camera ready.
>
> Overfitting of $\rho$.
> While the main point of the paper was to demonstrate the effect of $\rho$ on robustness, in order to clarify and clean up our argument, based on your question we have modified the results to be the ones achieved by cross validation.  I.e. we evaluate the robustness on a validation set, choose the rho that is best on the validation set and then report the test set performance, where the test set is the set of perturbations: Common Corruptions, Imagenet-A, and targeted PGD attacks.  We suggest using cross validation in practice and have emphasized this in the text of the paper as well.
>
> Questions about the Impact of BatchNorm.
> We agree that the empirical fact that BatchNorm seems to harm robustness of CIFAR10 models is intriguing. Note that we do not disable BatchNorm in the ImageNet models and still show real robustness improvements on the common corruptions benchmark as well as on targeted PGD attacks in the updated version of the paper.

---

### Official Review · AnonReviewer1 · 2019-10-23
**Official Blind Review #1**

**Rating:** 3

**Review:**

This paper studied the effectiveness of Conditional Entropy Bottleneck (CEB) on improving model robustness. Three tasks are considered to demonstrate its effectiveness; generalization performance over clean test images, adversarially perturbed images, and images corrupted by various synthetic noises. The experiment results demonstrated that CEB improves the model robustness on all considered tasks over the deterministic baseline and adversarially-trained classifiers.

The proposed idea is simple, easy to implement, and generally applicable to various classification networks. I especially enjoyed the nice experiment results; indeed, it has been widely observed that many existing approaches on adversarial training sacrifice the classification performance to improve the robustness over adversarial examples [1]. The proposed approach sidesteps such a problem since it does not require adversarial retraining. It is surprising to see that the proposed method is still able to achieve stronger robustness than the models based on adversarial training.

One of my major concerns is that it has a large overlap with Fischer (2018) in terms of methodology, experiment settings, and empirical observations, which limits the general contribution of the paper. Fischer (2018) first proposed CEB objective and showed its effectiveness in various tasks, including the adversarial defense. Although this paper extends the results on adversarial defense to CIFAR 10 dataset and includes additional ablative experiments and experiments on other types of input noises, it ends up confirming very similar observations/conclusions in Fischer (2018). Although Fischer (2018) is an unpublished work, I think that it is fair to consider that as a prior work since it is properly cited in the paper.

My other concern is that the experiment misses comparisons against other adversarial defense approaches, which makes it difficult to understand the degree of robustness this model can achieve. The current comparisons are mostly focused on deterministic and VIB baselines, which are useful to understand the core argument of the paper, but insufficient to understand how useful CEB could be in the purpose of adversarial defense. Especially, I believe that some recent approaches that do not require adversarial training, such as [A2], are worth comparisons.

Below are some minor comments/questions on the paper.
1. Section 3.2: For this experiment, BN is removed from the classification network; it would be still beneficial to see the baseline performance with BN (deterministic model) to better compare the classification performance on clean test data.
2. Section 3.3: The performance on both baseline and the proposed model on clean data is far lower than the SOTA models. Some clarification would be helpful.
3. It would be great to see further clarifications of improvement in CEB over VIB; currently, it is very simply justified that it is because CEB optimizes tighter variational bound on Eq.(3) than VIB. But it would also be great to see justifications from various angles (e.g. in the context of adversarial defense).


**Experience Assessment:**

I have published one or two papers in this area.

**Review Assessment: Checking Correctness Of Derivations And Theory:**

I assessed the sensibility of the derivations and theory.

**Review Assessment: Checking Correctness Of Experiments:**

I assessed the sensibility of the experiments.

**Review Assessment: Thoroughness In Paper Reading:**

I made a quick assessment of this paper.

---

> ### Author Response · Authors · 2019-11-15
> **Response to Reviewer1**
>
> Thank you for the detailed review. We respond to each of your concerns below.
>
> Overlap with Fischer (2018).
> We believe our paper offers novel contributions on top of Fischer (2018) and tried to fairly attribute it as necessary (despite it being unpublished work).  In particular, this paper offers a unique rederivation and justification of the CEB objective as well as a novel set of experiments.  In this work, we used a simpler invocation of the CEB objective and applied to larger datasets (Imagenet vs FashionMNIST).  Additionally, all of the Common Corruptions work is novel in comparison. The major contributions of this work are that the same technique scales easily to large classification problems like ImageNet and CIFAR10, and that the robustness conferred by training using the CEB objective applies to untargeted attacks and common corruptions as well. In our updated experiments, we are additionally showing robustness improvements on ImageNet models against ImageNet-A [1] and L$_2$ and L$_\infty$ targeted PGD attacks.
>
> Comparisons with other adversarial defenses.
> Thank you for the suggestion. In the context of this paper, we aim to demonstrate not that CEB is the best adversarial defense, but that it offers some substantial defense, which is interesting given that it is a general purpose objective function.  In principle this could be composed with other defense techniques and we agree that would make interesting follow up work. In particular, with regards to [A2], we'd like to emphasize that our CEB models do not show any decrease in their predictive performance and in fact show increased test accuracy in the regime in which they show adversarial robustness, a property that to our knowledge is unique.
>
> ImageNet Baselines.
> Due to feedback from the authors of [3] that the community is standardizing around reporting numbers on 224x224 Common Corruptions, in the updated draft we are switching from 299x299 Imagenet training and testing to 224x224, so the numbers in the new draft are slightly different. However, in both cases, our top-1 accuracies were well in-line with what we have seen reported in the literature for ResNet50, with and without AutoAugment. We imagine that the baselines could be improved by tuning learning rate schedules and weight decay factors carefully, but we don’t see any reason that similar tuning of the CEB models wouldn’t yield similar improvements. We have updated the appendix to mention that our implementation is closely based on the open-source Cloud TPU ResNet50 model from [2], and also to include a minimal set of modifications to that model to get the same basic CEB training setting that we used for the ResNet50 models without AutoAugment in this paper. We stuck with the ResNet models since many of our experimental results are meant to compare to other published Common Corruptions results, which are given largely for ResNet50 style models.  Are there particular architectures you were hoping to see compared instead?
>
> BatchNorm Baseline for CIFAR10.
> This is a good suggestion. We trained a deterministic 28x10 Wide Resnet model with BatchNorm and AutoAugment. It got 97.3% accuracy on the CIFAR10 test set, which is in-line with the 97.4% accuracy for that model reported in the AutoAugment paper. It got 0% accuracy on the baseline L$_\infty$ PGD attack at $\epsilon=8$ throughout training (we measured test robustness on L$_2$ and L$_\infty$ PGD attacks for all of our models every two training epochs). Interestingly, final L$_2$ PGD attack accuracy was noticeably better than the deterministic baseline without BatchNorm (73% vs. 66%), though still is much worse than the best CEB models on that attack (over 80% accuracy). We have added these results to the paper.
>
> Comparisons between VIB and CEB.
> We agree that a more complete comparison with VIB would be nice, but we note that we are already at the page limit. We would also like to make certain that the reviewer saw Section 3.1, and in particular the right two panels of Figure 1, where we directly compare CEB and VIB Fashion MNIST models at the same range of $\rho$ values against targeted PGD L$_2$ and L$_\infty$ attacks. In that setting, the CEB models dominate the VIB models at both general classification accuracy and robustness, even though the measured lower bounds on $I(X;Z)$ were almost identical for the VIB and CEB models at each value of $\rho$. We think this section provides the type of comparison with VIB that you are asking for in your review. Please tell us if we have misunderstood.
>
> [1] Hendrycks et al. “Natural Adversarial Examples”. 2019. https://arxiv.org/abs/1907.07174.
> [2] https://github.com/tensorflow/tpu/tree/master/models/official/resnet.
> [3] Yin et al. “A Fourier Perspective on Model Robustness in Computer Vision”. 2019. https://arxiv.org/abs/1906.08988.

---

> > ### Author Response · Authors · 2019-11-15
> > **Adding public comment for the missing reference in the review and reply**
> >
> > The reference [A2] in the review and reply above is:
> >
> > [A2] Zhang et al., Defending against Whitebox Adversarial Attacks via Randomized Discretization, In AISTATS, 2019.

---

### Author Response · Authors · 2019-11-15
**General Response**

We have given detailed replies to reviewer questions and concerns below. Based on reviewer feedback, we have additionally updated the paper as follows:
- We have added ImageNet results for ResNet 152 models, to show that the same trend of larger CEB models improving robustness over their deterministic baselines, with particular dramatic effect on the new PGD attacks.
- We have added targeted PGD L$_2$ and L$_\infty$ attacks to the ImageNet experiments, as well as results on Natural Adversarial Examples [1].
- We now point out in the Introduction the incidental but important contribution that the adversarially-trained CIFAR20 model of Madry (2017) failed to generalize to L$_2$ PGD attacks, having much worse performance than a deterministic baseline model of the same architecture on that attack.
- We have improved the legibility of Figures 1 through 3.
- We have added a deterministic baseline CIFAR10 model that uses BatchNorm to the comparison.
- We have added a code listing to the Appendix showing the core changes that are needed to turn the CloudTPU ResNet50 model [2] into a CEB model.
We hope that our discussion in the replies has helped the reviewers understand the paper in much greater detail, and that the changes listed above have adequately addressed reviewer’s comments. We look forward to answering any additional questions that reviewers may have in the remaining review period.

[1] Hendrycks et al. “Natural Adversarial Examples”. 2019. https://arxiv.org/abs/1907.07174.
[2] https://github.com/tensorflow/tpu/tree/master/models/official/resnet.

---

### Decision · Program_Chairs · 2019-12-19

**Decision:**

Reject

**Comment:**

This paper proposes CEB, Conditional Entropy Bottleneck, as a way to improves the robustness of a model against adversarial attacks and noisy data. The model is tested empirically using several experiments and various datasets.

We appreciate the authors for submitting the paper to ICLR and providing detailed responses to the reviewers' comments and concerns. After the initial reviews and rebuttal, we had extensive discussions to judge whether the contributions are clear and sufficient for publication. In particular, we discussed the overlap with a previous (arXiv) paper and decided that the overlap should not be considered because it is not published at a conference or journal. Plus the paper makes additional contributions.

However, reviewers in the end did not think the paper showed sufficient explanation and proof of why and how this model works, and whether this approach improves upon other state-of-the-art adversarial defense approaches.

Again, thank you for submitting to ICLR, and I hope to see an improved version in a future publication.